# Factors Associated with Knowledge, Attitudes, and Practice towards Colorectal Cancer and Its Screening among People Aged 50–75 Years

**DOI:** 10.3390/ijerph18084100

**Published:** 2021-04-13

**Authors:** Florence M. F. Wong

**Affiliations:** School of Nursing, Tung Wah College, Hong Kong, China; florencewong@twc.edu.hk; Tel.: +852-3468-6838

**Keywords:** attitudes, colorectal cancer screening, factors, knowledge, practice

## Abstract

Background: Colorectal cancer (CRC) screening is effective for early detection of CRC, particularly for males aged 50 or above. However, the rate of participation in the screening program is still low. This study was to examine knowledge, attitudes, and practice toward CRC and its screening and explored their associated factors. Methods: A descriptive cross-sectional study was conducted in a convenience sample of adults aged 50–75 years without cognitive problems, who were recruited at multi-elderly centers in Hong Kong. A questionnaire was used to measure knowledge, attitudes, and practice (KAP) towards CRC and its screening. Results: The total of 300 Chinese people included 147 (49.0%) males with a mean age of 58.72 (SD 6.91) years old. This study population had good knowledge and practice, as well as very good attitudes toward CRC and its screening. The multivariate regression results showed that receiving insurance coverage was the most significant factor positively associated with knowledge, attitudes, and practice. Other than this, lower educational level had significant negative association with knowledge and practice. Having self-sufficient financial support and receiving screening program information had positive associations with knowledge. Conclusion: People who are receiving insurance coverage have better KAP towards CRC and its screening. This indicates that they can receive adequate information about the screening procedure from their insurance agents and receive financial support under their insurance coverage. Therefore, they are more willing to participate in the screening program. Other factors, including having good self-sufficient financial support and receiving adequate information about CRC and its screening, significantly enhance knowledge. Based on the relationships among KAP, knowledge enhancement can improve attitude and practice in participating in the CRC screening program. Those who attained lower education should receive more attention. In this sense, adequate financial support from health insurance or subsidies from the government can increase an individual’s willingness to participate in the CRC screening, particularly those at a low socioeconomic level. Educational programs should be promoted to enhance knowledge about CRC and its screening, especially to those who attained lower education levels.

## 1. Introduction

### 1.1. Background

In 2018, colorectal cancer (CRC) was the third most common cancer worldwide, accounting for 1.80 million deaths [1]. In the USA alone, an estimated 27,150 men and 23,110 women died from CRC in 2017 [2]. CRC has been the second leading cause of cancer deaths in Hong Kong, accounting for 14.9% of all cancer deaths in 2017. The incidence rate of CRC increases with age, particularly in people aged 50 or above, accounting for more than 90% of 5036 newly diagnosed cases in Hong Kong [3]. Perhaps the most alarming point is that the disease is often asymptomatic, leading to late treatment and high mortality.

CRC screening presents an essential and cost-effective secondary prevention and disease control strategy to significantly increase the survival rate due to early detection and specific medical approaches [1,2], particularly for those who are asymptomatic. Therefore, attention to the utilization of CRC screening has increased for high-risk groups in multiple countries [4]. Due to increased risk of CRC beginning at 50 years old and in the interest of reducing the healthcare burden arising from CRC, a three-year government-subsidy screening program for Hong Kong residents aged 50 to 75 was launched in 2016 [5]. Similar CRC screening programs (CRCSP) have been promoted in other countries [6,7], however the participation rates have been low [5,6,7]. Evidence has shown that poor knowledge affects attitudes and practices of participation in CRC [8,9,10,11]. Studies were conducted to understand factors associated with participation rates in CRC screening [12,13]. However, their factors were diverse, and it was difficult to fully understand the low participation rates of CRC screening. Therefore, this study’s aims were to examine the status of KAP towards CRC and its screening and to identify factors associated with CRC screening among people aged 50 to 75.

### 1.2. Literature Review

CRC screening is currently a public health promotional scheme for early detection of CRC in asymptomatic populations [3]. The most common tests in CRC screening include the fecal occult blood test (FOBT), sigmoidoscopy, and colonoscopy. Virtual colonoscopy and stool DNA can be done to improve the accuracy in CRC screening [3]. Other new modalities used as screening tools, such as tomographic colonography and immunochemical tests, are useful in CRC screening [13,14]. Since 2016, the Hong Kong government has subsidized the CRC screening in two phases for Hong Kong residents aged 50 to 75. The screening tests start with the fecal immunochemical test (FIT) after the first medical consultation in the first phase. If the FIT result is positive, the eligible participant will receive a subsidy for further medical consultations and a standard colonoscopy service package with surgical treatment in the second phase. According to a report from the Department of Health in Hong Kong, about 30% (*n*=1423) of eligible persons enrolled in the screening program were found to be FIT-positive, and of these, more than 76.5% were found to have adenoma or adenocarcinoma [5]. Screening has been acknowledged as an effective detection tool for early diagnosis and treatment and has also reduced the burden of CRC healthcare costs [10,13,15].

Inadequate knowledge results in poor attitudes and participation in CRC screening [10,13]. A recent local study was conducted in a population aged 26 or older and found that fewer than 30% of subjects had adequate knowledge about CRC and its screening [13]. This resulted in a low participation rate in the CRCSP because they were not well-informed about the CRC or the screening tests for CRC and were concerned about possible discomfort during procedures and expenses of the program [13]. Moreover, attitudes towards CRC screening played an important role in the willingness of participation [15]. Identifying factors associated with KAP towards CRC screening have been the focus of recent studies [16,17,18,19].

In recent studies, numerous factors associated with poor participation were identified, including old age, low educational level, the recommendation of nurses or physicians, frequency of medical consultation, poor awareness of CRC screening, and inadequate CRC screening promotion [16,17]. In an older Chinese population, a low participation rate for CRCSP was due to a higher level of cues to action and lower perceived knowledge barriers and severity fears [18,19,20,21]. Nevertheless, health promotion was emphasized to raise public awareness and improve the participation rate [12]. To enhance understanding of KAP towards CRC and its screening, the capability, opportunity, motivation, and behavior (COM-B) model was used to guide this study. The COM-B Model was developed by Michie et al. [22] to describe the interactions among capability, opportunity, motivation, and behavior. Capability refers to an individual’s physical and psychological capacity to engage in the activity concerned. It embraces knowledge and skills. Opportunity refers to factors that surround the individual to make specific behavior possible. Motivation refers to cognitive processes including habitual processes, emotional responses, and analytic decision-making to strengthen and direct behavior. The directions of arrows indicate the potential influences of the components on each other. This COM-B model was used to guide the understanding of KAP of people aged 50 to 75 to make CRC screening more comprehensible. Figure 1 shows the COM-B model with KAP components.

## 2. Materials and Methods

### 2.1. Design

A descriptive cross-sectional design with convenience sampling was conducted in elderly centers in various districts. Recruitment participants included persons aged 50 to 75 and who were able to communicate in Chinese but excluded those who had cognitive problems. According to the population estimates by age groups of the Hong Kong Census and Statistics Department for 2018, there were an estimated 2.5 million residents of the population aged 50 to 75. Therefore, a sample size of at least 385 participants would be necessary to meet 95% confidence, a margin of error of 5%, and assumption of a population proportion at 0.5.

The eligible participants were first identified by the staff of the elderly centers. Then, the staff informed the research team about the participation schedule of the participants for data collection. On the day of data collection, the participants were approached and informed about the study purpose and procedures following the study information sheet. Then, they were requested to sign an informed consent form. Data were collected using one set of questionnaires, including a demographics form (including personal details and information about CRC and its screening) and the KAP questionnaire, through in-person interviews. The interviews lasted for approximately 20–30 min. Participants were assured that all data related to their personal information would be kept strictly confidential and anonymous.

### 2.2. Instrument: KAP Questionnaire Related to CRC and CRC Screening

#### 2.2.1. Validity and Reliability of the Instrument

To understand more about KAP with regards to CRC and its screening, a questionnaire was developed through three major steps, including a literature review for retrieval and modification of items and dimensions, review and comment by experts for accuracy and applicability of items and domains with content validity, and a pilot study for quantitative reliability.

The literature review was conducted to select the most relevant studies and related questionnaires were adopted as references [9,13,18,23] to develop a KAP questionnaire for the specific study population. The English version of the KAP questionnaire was developed as a draft and was also translated in Chinese and back-translated to English by a technical expert who was fluent in both languages. Then, the draft was reviewed by five experts for tool reliability and validity procedures. All experts were from clinical settings, including two medical physicians, two surgical doctors, and one senior nurse specialist who specialized in gastroenterology and hepatology. Their roles were to evaluate the accuracy and appropriateness of the items and domains on KAP in CRC and its screening. They were requested to rate each item from 1 to 4 for relevancy. A higher rating score indicates better relevancy of the item. Modification was made when narrative comments for a respective item were received. Content validity was conducted using quantitative analyses. The items were revised until the final content validity index reached 0.8, indicating good reliability. The finalized version was tested in a pilot study. The pilot study was carried out in a total 20 subjects for test–retest reliability and for evaluation of the flexibility and practicality of the study.

#### 2.2.2. Structure of the KAP Questionnaire

The questionnaire included three domains, namely knowledge, attitude, and practice (KAP) related to CRC and its screening. There were a total of 32 knowledge items, including two subdomains of knowledge related to CRC and eligibility of CRC screening, with “false” (score 0), “don’t know” (score 1), and “true” (score 2) choices. The score range was from 0 to 64. For knowledge related to CRC, there were three components, including signs and symptoms of CRC, CRC diagnostic tests, and CRC risk factors. There were seven items regarding attitudes and six items for practice towards CRC screening. Employing a 5-point Likert scale ranging from strongly disagree (score 1) to strongly agree (score 5), the score ranges for attitudes and practice were 7–35 and 6–30, respectively, whereby a higher score indicates better performance. The Cronbach’s alphas of knowledge, attitude, practice, and the overall scores were 0.86, 0.75, 0.70, and 0.86, demonstrating good to very good reliability.

### 2.3. Data Analysis

The SPSS v20.0 (IBM Corporation, Armonk, New York, USA) was used for data analysis. Descriptive statistics were applied to summarize and present the explanatory variables related to participants’ characteristics and the outcome variables (knowledge, attitudes, and practice (KAP)). Associations between the outcome variables (KAP) and demographic characteristics (age, gender, educational level, marital status, employment status, and others) were assessed by Chi-square or univariate analyses using Pearson’s correlation coefficient, independent-samples t-test, or one-way ANOVA (depending on the level of measurement of the outcome variables) to identify variables for multivariable regression. The variables with *p*-values < 0.25 were entered in the model for stepwise multivariable regression analysis to delineate factors independently associated with each outcome of KAP. The presence of collinearity was assessed by tolerance. Model adequacy was assessed by examining the scatter plot of standardized residuals against the predicted values and the normal probability plot of residuals. All statistical tests were two-sided and *p*-values < 0.05 were considered statistically significant.

### 2.4. Ethical Considerations

Approval was obtained from the research ethics committee of the study educational institute before the study started. Participants were requested to sign an informed consent form once they had agreed to participate in the study. They were assured that all data related to their personal information were kept strictly confidential and anonymous. Appendix A illustrates STROBE checklist for this cross-sectional study.

## 3. Results

### 3.1. Demographic Characteristics and CRC- and CRC-Screening-Related Information

A smaller sample size of a total of 300 Hong Kong Chinese participants aged 50 to 75 were recruited due to the time constraints of study period and the limited number of elderly centers available for participant recruitment and data collection. The mean age of the participants was 58.72 (SD 6.91) years old. Of the 300 participants, 49.0% (*n* = 147) were males, 84.0% (*n* = 252) were married or cohabiting, 68.7% (*n* = 206) had received secondary or higher educational level, 66.0% (*n* = 200) had an occupation, and 62.7% (*n*=188) were financially self-supported. Around half (50.3%, *n* = 151) had health insurance. More than 97% of the participants (*n* = 292) perceived themselves to have average or good health. Regarding CRC and CRC screening-related issues, more than 90% (*n* = 273) were not enrolled in the CRCSP. About 70% (*n* = 210) perceived themselves as having received sufficient information about CRC screening. Half of the participants (*n* = 128, ~50%) received CRC screening information from TV mass media. Most of the participants did not receive recommendations for the screening (74.3%). Only 2.7% (*n* = 9) had enrolled in the CRCSP. More than 80% (*n* = 267) could afford the CRC screening if it costed below HK$1,000. Table 1 shows the demographic characteristics and CRC- and CRC-screening-related details.

### 3.2. Knowledge, Attitudes, and Practice

The KAP means were 42.72 (SD 13.14), 23.52 (SD 3.70), and 16.81 (SD 3.45), respectively. The overall KAP mean was 78.83 (SD15.77). Significant correlations were found among KAP components (*p* < 0.001). The results of the associations among KAP components towards CRC screening are shown in Table 2.

### 3.3. Factors Associated with Participating in CRCSP

The multivariate regression results showed that having insurance coverage (regression coefficient (B) = 7.21, standard error of the regression coefficient (SE) = 1.68, *p* < 0.001; collinearity statistics: tolerance=0.90, Variance inflation factor (VIF) = 1.12), having self-support (B = 5.57, SE = 1.77, *p* = 0.002; collinearity statistics: tolerance =0.89, VIF = 1.12), and receiving CRCSP information (B = 13.12, SE = 2.32, *p* < 0.001; collinearity statistics: tolerance = 0.95, VIF = 1.05) were significantly positively associated with knowledge. However, having lower education (lower than primary education (B = −15.54, SE = 4.21, *p* < 0.001; collinearity statistics: tolerance = 0.92, VIF = 1.09) and primary education (B = −4.03, SE = 1.86, *p* = 0.031; collinearity statistics: tolerance =0.89, VIF = 1.12)) had significantly negative associations with knowledge (R2 = 0.289, F (5294) = 22.42, *p* < 0.001). The results also showed that having insurance coverage (B = 2.35, SE = 0.41, *p* < 0.001; collinearity statistics: tolerance =1.0, VIF = 1.0) had a significant positive association with attitudes (R2 = 0.38, F (3296) = 59.21, *p* < 0.001). Receiving insurance coverage (B = 2.01, SE = 0.38, *p* < 0.001; collinearity statistics: tolerance = 0.97, VIF = 1.04) had significant positive associations with practice but having primary education (B = −0.97, SE = 0.43, *p* = 0.025; collinearity statistics: tolerance = 0.97, VIF = 1.04) was significantly negatively associated with practice (R2 = 0.41, F (5294) = 40.45, *p* < 0.001). The results of multivariate regression analysis for factors associated with KAP towards CRC screening are summarized in Table 3.

## 4. Discussion

Considering the high prevalence and mortality rate of CRC worldwide and in Hong Kong, CRC screening is an effective tool for detecting CRC early, but it remains underutilized. In the present study, linear associations were found among KAP. The target population had fair KAP ratings, in which their attitudes were better than their knowledge and practice towards CRC screening.

The present study revealed the positive and negative factors associated with KAP, which may help explain the currently low participation rate of the CRCSP. Receiving insurance coverage was found to be the most significant factor associated with KAP of people aged 50 to 75 towards CRC screening. People who had insurance coverage for CRC screening had better knowledge, most probably because their insurance agent could explain the details about CRC screening and its procedures to them, meaning their perception of CRC and its screening was improved. Subsequently, they were more willing to participate in the CRCSP [8,10]. Insurance coverage reduces the psychological and financial burdens related to obtaining an expensive CRC screening test, leading to better attitudes and practices towards CRC and CRC screening [8,10]. Additionally, when people aged 50 to 75 receive financial support, such as insurance coverage, they will be more willing to receive education about CRC screening and participate in the CRCSP. Self-support is another significant factor found to be associated with knowledge towards CRC screening. Most of the target population aged 50 to 75 were retired, indicating that they no longer had an income. Therefore, having adequate financial support for the CRCSP is an important motivation to improve the KAP of people aged 50 to 75 towards CRC screening. The CRC screening includes different investigations, which the participants are required to pay for themselves. Since CRC screening includes several investigations, it may lay a heavy financial burden on the participants, leading to reluctance to participate, particularly for those at a lower socioeconomic level [10]. According to the current two-phase CRC screening subsidy scheme (of the two phases of CRC screening launched by the Hong Kong Government), people aged 50 to 75 will have FIT first, which determines the further medical consultation and standard colonoscopy service package with surgical treatment. Most of the people aged 50 to 75 (approximate 90%) would be willing to join the CRCSP if they only needed to spend $1000 Hong Kong dollars or less, and more than 50% of them were only able to pay at most $500 Hong Kong dollars. Evidence showed that Chinese people are usually financially conscious with regards to CRC screening, which they consider to be too expensive [24]. Previous local studies also found that Chinese people were usually unwilling to participate in the CRC screening unless the price was low or free [6,13]. Based on the COM-B model, an affordable fee can be the surrounding factor that affects opportunity and reduces motivation for joining CRCSP. Therefore, CRC screening can be a financial burden that holds the target group from understanding and participating in CRC screening unless the subsidy for the CRCSP from the HK government is adequate.

Subsequently, a lower educational level has been found to have significant negative associations with knowledge and practice towards CRC screening. People aged 50 to 75 who attained lower education (primary education or lower) had poorer knowledge related to CRC and CRC screening. It is understandable that those who have a lower educational level may need more support to understand the content of CRCSP. Additionally, those people are usually situated at a lower socioeconomic level and have more difficulties in obtaining adequate knowledge through new media. Based on the COM-B model, inadequate knowledge about CRC screening reduces capability, leading to poor motivation to participate in CRCSP. When the target population (age 50 to 75) do not receive adequate information, their intention or motivation to participate in CRCSP will be reduced [13,25]. Additionally, this study also found that providing information about CRCSP positively enhances the understanding about the program and importance of participation. However, while about 65% of participants received CRC screening information, their knowledge about CRC screening, such as symptoms of CRC, its risk factors, methods of CRC screening, and the importance of CRC screening, was still limited. It can be explained using the COM-B model that the motivation for participation in the CRCSP improves with the participants’ knowledge about it. The low participation rate for CRCSP may have been due to inadequate knowledge about CRC screening in the target group (eligible population aged 50–75). Evidence has shown that the Chinese population had poor attitudes towards CRC screening and was found to be relatively unaware of CRC screening, as knowledge about CRC screening was inadequate, leading to unwillingness to participate in the CRC screening [17,26,27]. Moreover, in Chinese culture, people may prefer delaying treatment of severe or fatal diseases to avoid emotional distress and financial difficulties [28]. Most of the at-risk Chinese population would receive the CRC screening only when they have symptoms. They do not want to bother or burden their families [24,28]. Despite this, more than 40% of target participants in this study received a recommendation for the CRCSP from their family members. This indicates that the family plays a vital role in providing support for their loved ones to participate in CRC screening [24].

### Strengths and Weaknesses

This study design prevented the detection of temporal changes in KAP and may not provide causal inferences between KAP and the associated factors. Therefore, longitudinal studies are recommended to detect changes across study periods. The present study results may not be generalizable to other age groups and non-Chinese societies, as only Chinese participants aged 50 to 75 were recruited due to cultural disparity and differences in healthcare service policies. Due to time constraints and limited elderly centers being available for subject recruitment, the inadequate sample size may have affected the reliability of the results.

## 5. Conclusions

Screening is an effective and successful method for reducing cancer morbidity and mortality. The present study can be treated as a follow-up study to provide new information about KAP towards CRC screening among the surveyed target group (aged 50 to 75) and to identify the associated positive and negative factors influencing their KAP. Identified factors are useful in suggesting improvements to the CRC screening program to achieve higher participation rates. Both the government and healthcare policy makers need to collaborate to improve awareness of and participation in CRC screening by enhancing education for the eligible population through various formats and promotional channels with familial involvement. The government may also offer more subsidies for CRC screening to the eligible population, particularly those at a lower socioeconomic level.

## Figures and Tables

**Figure 1 ijerph-18-04100-f001:**
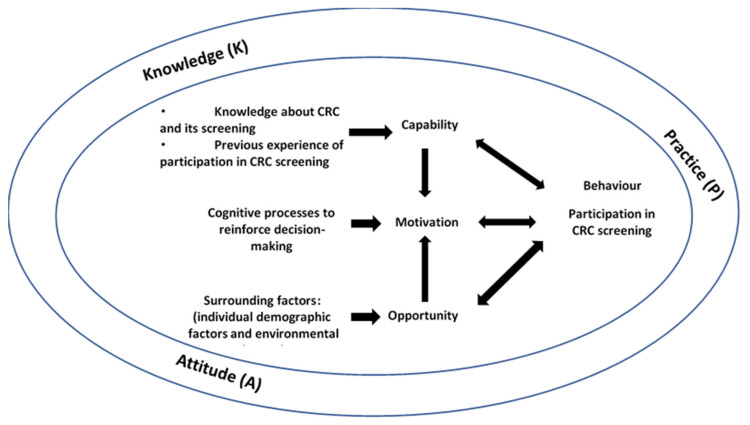
The COM-B model and KAP.

**Table 1 ijerph-18-04100-t001:** Demographics and general CRCSP-related information (*n* = 300).

Demographics	*n* (%)
Gender	
Male Female	147 (49%) 153 (51%)
Age	
50–55 56–60 61–75	121 (40.3%) 73 (24.3%) 106 (35.3%)
Marital status	
Single Married Divorced Widowed Co-habituated	13 (4.3%) 251 (83.7%) 20 (6.7%) 15 (5%) 1 (0.3%)
Educational level	
Illiterate Primary Secondary Tertiary	12 (4%) 82 (27.3%) 129 (43%) 77 (25.7%)
Employment status	
Employed (full-time) Employed (part-time) Self-employed Unemployed Retired Housewife/Househusband	156 (52%) 26 (8.7%) 18 (6%) 26 (8.7%) 45 (15%) 29 (9.7%)
Financial support	
Self-supported Supported by family Supported by government Others	188 (62.7%) 91(30.3%) 17 (5.7%) 4 (1.3%)
Health insurance coverage	
Yes No	151 (50.3%) 149 (49.7%)
Self-perceived health status	
Poor Average Good	8 (2.7%) 192 (64%) 100 (33.3%)
Relatives diagnosed with CRC	
Yes No	60 (20%) 240(80%)
Acknowledged with CRCSP	
Yes No	258 (86%) 42 (14%)
Source of acknowledgment	
TV Radio Friends Health care professional	209 (59.4%) 50 (14.2%) 59 (16.8%) 34 (9.7%)
Source of information	
Pamphlets Booklets Posters Educational video TV advertisement Radio Online information Newspaper Magazine	38 (11.7%) 22 (6.7%) 18 (5.5%) 12 (3.7%) 148 (45.4%) 29 (8.9%) 26 (8%) 26 (8%) 7 (2.1%)
Received information of CRCSP	
Yes No	197 (65.7%) 103 (34.3%)
Perceiving sufficient information	
Yes No	210 (70%) 90 (30%)
Recommended for CRCSP	
Yes No	77 (25.7%) 233 (74.3%)
Had received CRCSP information from	
Family member Friends Health care professional	41(43.6%) 29(30.9%) 24(25.5%)
Willingness of expenditure for CRCSP	
<$100 <$500 ≤$1000 >$1000	82(27.3%) 90(30%) 95(31.7%) 33(11%)

CRC: colorectal cancer; CRCSP: colorectal cancer screening program.

**Table 2 ijerph-18-04100-t002:** Correlations among KAP components towards CRC and its screening (*n* = 300).

	Correlation Coefficient			Significance (*p*)
	Knowledge (K)	Attitude (A)	Practice (P)	
Knowledge	-	0.56	0.59	<0.001 ***
Attitudes	0.56	-	0.56	<0.001 ***
Practice	0.56	0.59	-	<0.001 ***

Note: *** *p* < 0.001.

**Table 3 ijerph-18-04100-t003:** Results of multivariate regression for factors associated with KAP towards colorectal cancer screening.

	Knowledge			Attitude			Practice		
	B	SE	*p*	B	SE	*p*	B	SE	*p*
Lower than primary education	–15.54	4.21	<0.001 ***						
Primary education	–4.03	1.86	0.031*				–0.97	0.43	0.025 *
Having insurance coverage	7.21	1.68	<0.001 ***	2.35	0.41	<0.001 ***	2.01	0.38	<0.001 ***
Having self-support	5.57	1.77	0.002 **						
Receiving CRCSP information	13.12	2.32	<0.001 ***						

B: the regression coefficient; SE: standard error of the regression coefficient; *p*: significance: * *p* < 0.05; ** *p* < 0.01; *** *p* < 0.001.

## Data Availability

The data presented in this study are available on request from the corresponding author. The data are not publicly available due to privacy reason.

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
