# Peer review of "Factors Associated with Knowledge, Attitudes, and Practice towards Colorectal Cancer and Its Screening among People Aged 50–75 Years"

_ijerph, 2021, doi:10.3390/ijerph18084100_

Round 1

Reviewer 1 Report

INTRODUCTION: Overall this section is too long and can certainly be pared down. Section 1.1 seems to end with the study objective but then it is reiterates in 1.3 again. Interestingly, while the main focus of this manuscript is on KAP. The description of the theoretical model is included at the end of the background section yet labeled as 2.1. It is unclear as  to where it goes as it reads very technical and not placed within the context of the background or Methods.  The aim also is very specific about the details of the population. That should go in the Methods section.

METHODS: More information is required with regards to the sample population. Why was there a focus on "caregivers in oral care for institutionalized elderly"? Were all institutions sampled? How were people recruited? What was the participation level?  The analytic approach is also lacking. Why did the relationship between demographic variables and KAP include correlations coefficients? Collinearity among which variables were assessed?  I assume they used linear regression although they did not specify which model was used. Also the authors state that they "sought" IRB approval. Did they receive it? 

RESULTS: Section 3.1 does not read well. I got lost in reading it. It can be simplified. I also could not identify where all the numbers came from. Also Table 1. should exclude Age mean as that is already included in the Results narrative and is the only mean in the whole table. Thus the table can be simplified in terms of headings and footnotes.  Also CRSP was not spelled out in the table.  Table 2 is not needed as the data is provided in the narrative although it should be clear as to what means belong to knowledge versus attitudes versus practice.  If Table 2 is left in then at least add ranges.  I do not understand the purpose of Table 3. 

DISCUSSION: This section was generally fine and appropriate although the first paragraph can be reduced to a sentence and lead into a brief summary of the main findings. 

Throughout the paper one kind fine grammatical errors and sentences that do not quite make sense. It deserves a closer review and edit. 

Author Response

Thank you very much for your valuable comments.

Please find the responses below. I hope that you are satisfied with the revised manuscript for publication.

Reviewer 2 Report

The paper uses 300 sample data to study the knowledge, attitudes and practices of CRC for people aged 50-75, and gives a demographic analysis and impact analysis.

The paper can be used to guide the early detection of CRC and has scientific significance.

The questionnaire design is rigorous, the statistical analysis is scientific and reasonable, and the experimental results are credible.

There are some small problems:

  1. The scientific basis for a sample size of 300?

Is the sample size sufficient?

  1. Are all 300 samples valid?

Is there any failure of the sample? 

Author Response

(The authors gave the same response as above.)

Reviewer 3 Report

Wong investigated the knowledge, attitude and practice in people of 50- 75 years old towards colorectal cancer and the screening of colorectal cancer in Hong Kong and found multiple factors associated with it which affects participation rate.

Major concern:

The author describes the development and validation of a self-developed KAP questionnaire however, information provided in paragraphs 2.2 and 2.3 is identical and overlapping regarding the development of the KAP questionnaire. Furthermore, development and validation of this questionnaire has not been mentioned as a study aim. Lastly, if the author developed such a tool, more extensive information should be provided about the development and validation process to assure its validity. Information regarding the development of the tool is lacking and therefore there are major concerns about its validity.

Additionally, if the author developed the tool, a power calculation is needed to show that the number of participations is sufficient to validate and test the tool.

Finally, when describing the methods or results, the author does not mention what the elements were that are in the KAP questionnaire. It is therefore unclear how to interpret the results. Because the questionnaire has not been used before and was developed for this study, no prior publication can be referenced. I therefore strongly suggest to add the information regarding development and validation of the questionnaire to the current manuscript. It may be preferable however to first publish a paper on the validation of the questionnaire, which can be referenced in this paper as background article.

Abstract:

  • It is unclear to the reader with the information in the sentence “Numerous factors had remarkably association with knowledge [R2 = 0.289, F(5,294) = 22.42, P<0.001, attitudes [R2= 0.38, F (3.296)= 59.21, P< 0.001] and practice [R2= 0.41 F(5,294)= 40.45, P 0.001]” means as information regarding the questionnaire has not yet been provided.

Introduction:

  • The meaning of the following sentence is unclear: “Virtual colonoscopy and stool DNA can be emerged in CRC”
  • Does the author mean “adequate” knowledge to underline the lack of knowledge in the following sentence: “fewer than 30% had inadequate knowledge about CRC”
  • The aim described in paragraph 1.3 do not overlap with the information provided in title and abstract. So far, it had seemed as if the author was examining the KAP among patients 50-75 years towards CRC and CRC screening however here the author states that the aim of the study was to develop a tool to measure the KAP in oral care!!!

Methods

  • The information of paragraph 2.1 should be implemented in the introduction. Currently, there are 2 paragraphs named 2.1. The meaning of this paragraph in current manuscript is unclear.
  • Please add information regarding the obtained approval of the research ethics committee to paragraph 2.4 as unclear if approval was received.

Discussion:

  • Please revise the following sentence: “It indicates that financial support is concerned the most by population aged 50 to 75 in participating in CRC screening programme.”

Minor comments:

Title:

  • Consider removing “among people aged 50-75” from the title or add “years”

Abstract:

  • Add “a” before “convenience sample”
  • Abstract: add “years” following “50-75”

Introduction:

  • Remove “of” in from in “1.80 million of deaths”
  • Consider adding “secondary” in front of “prevention” in the second paragraph
  • Replace “aged” by age in second paragraph
  • Make “similar CRC screening programme has” plural as the author is referring to multiple programs.
  • Replace “their” with “these” at the end of paragraph 1.1
  • Replace “be” in “have” in the following sentence “were found to be adenoma or adenocarcinoma”

Table 1:

  • Remove mean/ SD from the top row as this is not used in the table.
  • Explain CRCSP at first use

Author Response

(The authors gave the same response as above.)

Round 2

Reviewer 3 Report

Dear author,

Thank you for submitting a revised version of the manuscript following the comments. 

After reading the revision, I suggest to move the study aim (currently paragraph 2.1) from the methods section to the introduction (last paragraph of introduction).

Author Response

After reading the revision, I suggest to move the study aim (currently paragraph 2.1) from the methods section to the introduction (last paragraph of introduction).

Response: Thank you for your advice and support. The study aim has been moved to the last paragraph of introduction.
